# Study on the Performance Mechanism of Polyformaldehyde Glycol Ether Polymer for Crude Oil Recovery Enhancement

**DOI:** 10.3390/ma17020437

**Published:** 2024-01-16

**Authors:** Shaohui Jiang, Wenxue Lu, Tao Li, Fujun Ma, Dahu Yao, Qingsong Li

**Affiliations:** 1The State Key Laboratory of Heavy Oil Processing, China University of Petroleum East China, Qingdao 266580, China; shaohuijiang79318@163.com; 2Shandong Energy Group Co., Ltd., Jinan 250100, China; wenxuelu5599@163.com (W.L.); taoli197808@126.com (T.L.); 3School of Chemistry & Chemical Engineering, Henan University of Science and Technology, Luoyang 471023, China; fujun998877@126.com (F.M.); dahu45678@126.com (D.Y.)

**Keywords:** polyformaldehyde glycol ether polymers, surfactant, surface tension, interfacial tension, oil washing efficiency, drainage efficiency

## Abstract

The demand for energy continues to increase as the global economy continues to grow. The role of oilfield chemicals in the process of oil and gas exploration, development, and production is becoming more and more important, and the demand is rising year by year. The support of national policies and the formulation of environmental protection regulations have put forward higher requirements for oilfield chemical products, which has promoted the innovative research and development and market application of oilfield chemicals. Polyformaldehyde glycol ether polymer (PGEP) is simple to synthesize, easily biodegradable, green and environmentally friendly, and in line with the development trend of chemicals used in oil and gas development. The interfacial tension performance of PGEP after compounding with different surfactants can reach as low as 0.00034 mN/m, which meets the requirements of the oilfield (interfacial tension ≤ 5 × 10^−3^ mN/m). The best oil washing efficiency performance of PGEP compounded with different surfactants reached 78.2%, which meets the requirements of the oilfield (oil washing efficiency ≥ 40%). The fracturing fluid drainage efficiency of PGEP after compounding with different surfactants reaches 22%, which meets the requirements of the oilfield (drainage efficiency ≥ 15%). The surface interfacial tension of the system remains constant after the concentration exceeds 0.2% and decreases with lower concentrations. The drainage efficiency increases with increasing concentrations in the range below 0.6%. It was determined that PGEP can be used as a surfactant instead of fatty-alcohol ethoxylates (FAE) in oilfield development.

## 1. Introduction

Demand for oilfield chemicals has risen rapidly in recent years. Over the past decade, a trillion barrels of global crude oil are expected to have been produced because of the use of oilfield chemicals [1,2,3,4,5,6]. It is expected that the consumption of oilfield chemicals will increase at an average annual rate of 10%, and the number of varieties has now reached more than 300. Oil and gas development uses the largest amount and the largest category of chemicals [7,8,9,10,11,12,13,14]. The various types of chemicals currently used have various problems in their production, such as high cost, environmental pollution, poor results, etc. There is an urgent need to develop new green and environmentally friendly chemicals for oil and gas development. FAE has good detergent and emulsifying properties [15], insensitivity to water hardness, high biodegradability [16], low toxicity compared to alkylphenolethoxylates (APE) [17], and a low production cost compared with alkyl polyglucosides (APG). In addition, they can be broadly used as co-surfactants due to their compatibility with other surfactants, such as sodium lauryl ether sulfate, cocoamidopropyl betaine, or sodium lauroyl sarcosinate [18,19,20,21]. FAE is one of the most important groups of nonionic surfactants [22,23,24,25]. The ether bond in the molecule is not easy to destroy by acid and alkali, so it has high stability, good water solubility [26,27,28,29,30], electrolyte resistance, easy biodegradation, and small foam [31,32,33,34,35,36,37,38]. This type of surfactant is produced by an addition reaction between fatty alcohols and ethylene oxide, and there is a risk of environmental contamination during the synthesis process [39]. Ethylene oxide is a toxic carcinogen and belongs to class 1 carcinogens. Ethylene oxide is flammable and explosive and is not easy to transport over long distances [40]. Therefore, the synthesis process is dangerous and causes greater pollution to the environment.

In this paper, PGEP was synthesized, which is simple to synthesize, high yield, low cost, biodegradable, green, and environmentally friendly, which is in line with the development trend of chemicals used in oil and gas development. The synthesis of PGEP is based on paraformaldehyde and polyol as raw materials. The synthesis process is simple, easy to biodegrade, and the degraded products are basically alcohols, which do not easily cause environmental pollution. Therefore, the product was synthesized, evaluated, and experimented with in terms of its performance with the expectation of replacing FAE in oilfield applications [30,37]. In this paper, the study of PGEP enhanced crude oil recovery by compounding PGEP with different cationic, anionic, nonionic surfactants, and alcohols through oil washing efficiency and drainage efficiency [31,32,33,34,35,36].

## 2. Materials and Methods

### 2.1. Materials and Instruments

Methanol and Ethanol (analytically pure) were purchased from Tianjin Kermel Chemical Co., Ltd., Tianjin, China. Ethylene Glycol (analytically pure) was purchased from Tianjin Best Chemical Co., Ltd., Tianjin, China. Isopropyl Alcohol (analytically pure) was purchased from Tianjin Kaitong Chemical Co., Ltd., Tianjin, China. Kerosene (analytically pure) was obtained from the Tianjin Damao Chemical Factory, Tianjin, China. 

Crude oil samples (B33H, LD5-2, Q104, CL55) were obtained from various fields.

1231 (Dodecyl trimethyl ammonium chloride, industrial product, 30% content) was obtained from Shandong Anmi Chemical Technology Co., Ltd., Zibo, China. 1631 (Cetyltrimethylammonium chloride, industrial product, 30% content) was obtained from Huainan Huajun New Material Technology Co., Ltd., Huainan, China. 1831 (Octadecyl trimethyl ammonium chloride, industrial product, 30% content) was obtained from Huainan Huajun New Material Technology Co., Ltd. BS12 (Dodecyl trimethyl betaine, industrial product, 30% content) was obtained from Shandong Anmi Chemical Technology Co., Ltd. LHSB (Dodecylamino hydroxy sulfobetaine, industrial product, 30% content) was obtained from Zhejiang Xinsheng Oil & Grease Technology Co., Ltd., Jiaxing, China. CAB35 (Cocamidopropyl betaine, industrial product, 35% content) was obtained from Huainan Huajun New Material Technology Co., Ltd. LAB30 (Lauramidopropyl betaine, industrial product, 35% content) was obtained from Huainan Huajun New Material Technology Co., Ltd. OAB40 (Oleic acid amidopropyl betaine, industrial product, 40% content) was obtained from Huainan Huajun New Material Technology Co., Ltd. AES (Sodium Fatty Alcohol Polyoxyethylene Ether Sulfate, industrial product, 70% content) was obtained from Jiangsu Shengtai Science and Technology Co., Ltd., Taizhou, China. K12 (sodium dodecyl sulfate, industrial product, 90% content) was obtained from Shandong Jujin Chemical Co., Ltd., Zibo, China. AOS (sodium allyl sulfonate, industrial product, 30% content) was obtained from Shandong Jujin Chemical Co., Ltd. APG (Alkyl Glycosides, industrial product, 50% content) was obtained from Shanghai Fakai Chemical Co., Ltd., Shanghai, China. CPS30 (Coconut oil fatty acid potassium soap, industrial product, 90% content) was obtained from Hubei Keward Chemical Co., Ltd., Wuhan, China. NPES (nonylphenol polyoxyethylene ether sodium sulfate, industrial product, 90% content) was obtained from Jiangsu Hai’an Petrochemical Factory, Nantong, China. MOA-07 (Fatty alcohol polyoxyethylene ether, industrial product, 90% content) was obtained from Nantong Chenrun Chemical Co., Ltd., Nantong, China. OP-10 (polyoxyethylene octyl phenol ether-10, industrial product, 99% content) was obtained from Shandong Wantai Chemical Co., Ltd., Jinan, China.

Surface interfacial tensiometer (k100c), Kruss, Hamburg, Germany. Rotational Ultra-Low Interfacial Tension Meter (TX500C), Texas Instruments, Dallas, TX, USA. Digital High-Speed Mixer (GJ-3S), Qingdao Haitongda Special Instrument Factory, Qingdao, China. Constant temperature water bath (HH-6), Changzhou Guohua Electric Co., Ltd., Changzhou, China. Electronic Balance (AL204), Mettler Toledo Instruments Co., Ltd., Zurich, Switzerland. Thermostatic drying oven (OMH100-S), Thermo Fisher Scientific, Waltham, MA, USA.

### 2.2. PGEP Prepared

The polyformaldehyde monomer and ethylene glycol monomer were reacted by an acid catalyst with stirring at a reaction temperature of 120 °C and a reaction pressure of 0.3–0.5 MPa for 8 h. At the end of the reaction, soda ash was added to neutralize the acid catalyst, and the reaction was stirred at room temperature for 1 h. The solids obtained by filtration were dried and processed at 100 °C for 20 h, and PGEP with stable performance was obtained.

### 2.3. Surface Tension Test Methods

The sample container was moved to bring the plate into contact with the liquid. Recorded the current reading m. The plate was in contact with the liquid, recorded the Wetted portion of the periphery of the plate l. Surface tension was calculated as follows:(1)$$\sigma =\frac{m\times g}{l}$$

m—mass of liquid pulled up by the plate, g; g—gravitational acceleration, cm/s^2^; l—Wetted portion of the periphery of the plate, twice the length of the plate plus twice its thickness, cm.

### 2.4. Interfacial Tension Test Method

The low-density phase liquid was drawn up with a microsyringe and injected into the measuring tube to form a droplet. The interface tensiometer TX500C was started, and the rotational speed was adjusted so that the ratio of the droplet length L to the droplet diameter d in the measuring tube was as close as possible to 2–8. Three consecutive measurements with a size difference of less than 0.1 were considered steady state. Interfacial tension was calculated as follows:(2)$$\sigma =\left(1/8\right)\times \pi^{2}\cdot ∆\rho \times 10^{6}\cdot \left(d^{3}/m^{3}\right)\cdot \left(1/n^{3}\right)\cdot \left(1/n^{3}\right)\cdot \left(1/R^{2}\right)\cdot f\left(L/d\right)$$

σ—interfacial tension, mN/m; ∆ρ—difference in density between the two phases, g/cm^3^; R—instrument panel rotation speed, ms/r; n—external refractive index; d—droplet diameter, cm; L—droplet length, cm; m—instrument microscope magnification; $f\left(L/d\right)$—calibration factor.

### 2.5. Oil Washing Efficiency Test Method 

The simulated stratum sand was mixed with the target block crude oil at a ratio of 4:1 (mass ratio) and aged at the reservoir temperature for 7 days. A 0.5% sample solution was prepared from the injected water of the target block. Aged oil sand was placed into a 100 mL conical flask and weighed to obtain $m_{1}$. The prepared sample solution was added to the aged oil sand sample and allowed to stand for 48 h at reservoir temperature. The floating crude oil in the sample solution and the crude oil adhering to the wall of the bottle after resting were dipped out with clean cotton gauze. The conical flask was baked in an oven at 105 °C until a constant weight was achieved and $m_{2}$ was obtained. The sample was eluted from the crude oil with petroleum ether until the petroleum ether was colorless. The conical flask with eluted crude oil was baked in an oven at 120 °C until a constant weight was achieved and weighed to obtain $m_{3}$. Oil washing efficiency was calculated as follows:(3)$$\eta =\frac{m_{1}-m_{2}}{m_{1}-m_{3}}\times 100\%$$

η—Oil washing efficiency, %; $m_{1}$—Total mass of conical flask and oil sand before oil washing, g; $m_{2}$—Total mass of conical flask and oil sand after oil washing, g; $m_{3}$—Total mass of conical flask and washed formation sand, g.

### 2.6. Drainage Efficiency Test Method

The glass tube with an inner diameter of 15 mm and a length of 500 mm was filled with quartz sand with a grain size of 0.180–0.280 mm and fitted according to the process. The height of the liquid level was controlled to maintain a constant pressure head. It was saturated by passing a 2% aqueous solution of KCl, and the pore volume V was found from the mass difference before and after saturation. Kerosene was passed positively into the sand-filled tube, and the amount of KCl aqueous solution discharged at the start of the kerosene flow is recorded $Q_{1}$. The 2% KCl aqueous solution was back-passed into the sand-filled tube. Recorded the amount of kerosene discharged $Q_{2}$ when it began to flow out. Then kerosene was passed positively into the sand-filled tube, and the discharge of the KCl aqueous solution $Q_{3}$ was recorded when the kerosene started to flow out. Discharge efficiency was calculated as follows:(4)$$B_{0}=\frac{Q_{3}}{\left(V-Q_{1}+Q_{2}\right)}\times 100\%$$$B_{0}$—discharge efficiency, %; $\left(V-Q_{1}+Q_{2}\right)$—volume of liquid in the sand filling tube, mL; $Q_{3}$—volume of liquid discharged from the sand filling tube, mL.

The 2% KCl aqueous solution was added with a 0.3% drainage agent, and the test steps were repeated. Drainage efficiency was calculated as follows:(5)$$E=\frac{B-B_{0}}{B_{0}}\times 100\%$$E—drainage efficiency, %; $B_{0}$—discharge efficiency of blank specimens, %; B—discharge efficiency of specimens with drainage agent.

## 3. Result and Disscusion

### 3.1. PGEP Infrared Spectral Characterization and Analysis

Figure 1 shows that the absorption peak at 2935.66 frequency is the asymmetric stretching vibration of paraformaldehyde CH_2_, the absorption peak at 2877.79 frequency is the symmetric stretching vibration of paraformaldehyde CH_2_, the bending stretching vibration of paraformaldehyde CH_2_ at 1458.18, and the C-O stretching vibration of paraformaldehyde at 1284.59, 1037.70, and 848.68, 3495.01, and 1365.60 indicate that the hydroxyl group in the molecular chain is the primary hydroxyl group.

The C-O-C stretching vibration peak at 1118.71 indicates the presence of aliphatic ether in the molecular chain, and it can be seen that the PGEP with the indication of activity was formed in this reaction.

### 3.2. Study on PGEP Surface Tension Performance 

The lowest surface tension of PGEP surfactant can reach 27.4 mN/m, as shown in Figure 2.

The surface activity of FAE is higher than that of common hydrocarbon surfactants, with a critical micelle concentration of 0.01%.

### 3.3. Study on the Interfacial Tension Performance of PGEP after Compounding with Different Surfactants

#### 3.3.1. Effect of Different Components in Cationic Systems on Interfacial Tension

Experimental method: 3.5 g of PGEP, 3.5 g of cationic surfactant, 9 g of alcohol, 17 g of amphoteric surfactant, and 67 g of water were taken, and the interfacial tension was measured after complete dissolution.

Figure 3a indicates that 1631 has the best interfacial tension, followed by 1231, but there is little difference between the two surfactants.

1831 has the weakest interfacial tension of crude oil and differs from both by almost an order of magnitude. Figure 3b shows that the interfacial activities of these four betaines behaved differently for different crude oil samples. For the B33H crude oil sample, CAB35 has the lowest interfacial tension. For the LD5-2 crude oil sample, LAB30 has the lowest interfacial tension. For the Q104 crude oil sample, none of them can form ultra-low interfacial tension. For the CL55 crude oil sample, BS12 has the lowest interfacial tension. Betaine has an impact on interfacial tension, which is inconsistent across crude oils. However, the interfacial tensions are all in the same order of magnitude and have little effect. Figure 3c shows that ethanol is the best of the alcohols, followed by methanol, and then isopropanol. The poorest performing is ethylene glycol. Except for ethylene glycol, where the interfacial tension is an order of magnitude higher, there is little difference in the interfacial tensions of the other three alcohols.

For cationic surfactant systems, the interfacial tension of the system depends on the type of cationic surfactant, with 1231 performing best. Betaine affects interfacial tension within an order of magnitude. Ethylene glycol decreases the interfacial activity of the system. Other alcohols do not affect the interfacial tension of the system.

#### 3.3.2. Effect of Different Components in Anionic Systems on Interfacial Tension

Experimental method: 3.5 g of PGEP, 3.5 g of anionic surfactant, 9 g of alcohol, 10 g of amphoteric surfactant, and 74 g of water were taken, and their performance were measured after complete dissolution.

Figure 4a illustrates that there is no clear pattern in the interfacial tension behavior of anions for different crude oil samples.

For the B33H crude oil sample, the AES interfacial tension was the lowest, but for the CL55 crude oil sample, k12 had the lowest interfacial tension. The interfacial tensions of the same crude oil are all in the same order of magnitude. This indicates that the anionic surfactant has little effect on the interfacial tension of the crude oil. Figure 4b indicates that the different amphoteric surfactants have little effect on the interfacial tension of crude oil under the condition that the anion is kept constant and the crude oil sample is kept constant. The effect of alcohols is the same as for the cationic system, as shown in Figure 4c. Methanol, ethanol, and isopropanol had little effect on the change in interfacial tension of the system, except for ethylene glycol, which resulted in a large change in interfacial tension.

For anionic systems, the effect of different anionic and amphoteric surfactants on interfacial tension is within an order of magnitude. The addition of ethylene glycol leads to a decrease in interfacial activity.

#### 3.3.3. Effect of Different Components in Nonionic Systems on Interfacial Tension 

##### Effect of Different Nonionic Surfactants on Interfacial Tension

Experimental method: 3.5 g of PGEP, 3.5 g of nonionic surfactant, 9 g of alcohol, 10 g of amphoteric surfactant, and 74 g of water were taken, and the interfacial tension was measured after complete dissolution.

Figure 5a shows that the three nonionic systems exhibit different interfacial tensions for different crude oil samples.

For the B33H crude oil sample, OP-10 has the lowest interfacial tension. MOA-7 has the lowest interfacial tension for LD5-2 and Q104 crude oil samples, and APG has the lowest interfacial tension for CL55 crude oil samples. Figure 5b illustrates that for the nonionic system, CAB35 is the most effective and BS12 performs similarly. The other two interfacial tensions are on the high side. Figure 5c shows that for the nonionic system, the ethanol system has the lowest interfacial tension. The interfacial activity of all three systems, methanol, ethanol, and isopropanol, was excellent, except for ethylene glycol’s apparent poor interfacial activity.

For nonionic systems, the interfacial tension of different nonionic systems is related to the crude oil sample, which is more targeted. The effect of betaine type on interfacial tension was not more than of the same order of magnitude. Ethylene glycol, on the other hand, reduces the interfacial activity of the system, and the interfacial tension changes by more than an order of magnitude.

### 3.4. Study on the Performance of PGEP Compounded with Different Surfactants on Oil Washing Efficiency

#### 3.4.1. Effect of Different Components in Cationic Systems on Oil Washing Efficiency

Experimental method: 3.5 g of PGEP, 3.5 g of cationic surfactant, 9 g of alcohol, 17 g of amphoteric surfactant, and 67 g of water were taken, and their oil washing efficiency was measured after complete dissolution.

Figure 6a indicates that the oil washing efficiency varies significantly, with 1231 having the highest oil washing efficiency.

Both the 1631 and 1831 systems have lower oil washing efficiency. Figure 6b shows that for the cationic system, CAB35 has the highest oil washing efficiency, followed by BS12, with a small difference in the oil washing efficiency. Figure 6c indicates that ethylene glycol has the lowest oil washing efficiency. However, the difference in the oil washing efficiency of the four alcohols is not significant, indicating that the effect of alcohols is small.

The oil washing efficiency of the cationic system is strongly influenced by both cationic surfactant and amphoteric surfactant. 1231 is the best of the cationic surfactants, and CAB35 and BS12 are the best of the amphoteric surfactants.

#### 3.4.2. Effect of Different Components in Anionic Systems on Oil Washing Efficiency 

Experimental method: 3.5 g of PGEP, 3.5 g of anionic surfactant, 9 g of alcohol, 10 g of amphoteric surfactant, and 74 g of water were taken, and their oil washing efficiency was determined after complete dissolution.

Figure 7a shows that for the anionic surfactant system, K12 has the highest oil washing efficiency, followed by CPS30 and AOS.

There is not much difference in oil washing efficiency between CPS30 and AOS. NPES has the lowest oil washing efficiency. Figure 7b shows that among the anionic systems, BS12 had the best oil washing efficiency. The oil washing efficiency of the other betaine surfactant systems did not differ much except for OAB40, which had a lower oil washing efficiency. Figure 7c shows that in the anionic system, different alcohols have little impact on the oil washing efficiency, which is almost the same. It indicates that alcohols have no effect on the oil washing efficiency of the anionic system.

The oil washing efficiency of the anionic system is most affected by the anionic surfactant, with a maximum difference of more than 10%. K12 has the best oil washing efficiency. Amphoteric surfactant systems had little effect on oil washing efficiency, except for OAB40 (a reduction of more than 5%). Alcohols have little effect on the oil washing efficiency of the system.

#### 3.4.3. Effect of Different Components in a Nonionic System on Oil Washing Efficiency

Experimental method: 3.5 g of PGEP, 3.5 g of nonionic surfactant, 9 g of alcohol, 10 g of amphoteric surfactant, and 74 g of water were taken, and their oil washing efficiency was measured after complete dissolution.

Figure 8a shows that OP-10 has the best oil washing efficiency for the nonionic system, with MOA-7 coming in a close second.

APG has the poorest oil washing efficiency. Figure 8b shows that for the nonionic system, BS12 has the highest oil washing efficiency. For different crude oil samples, the oil washing abilities of OAB40, CAB35, and BS12 were not consistent with each other. Figure 8c indicates that in the nonionic system, alcohols do not have a significant effect on the oil washing efficiency. In comparison, ethylene glycol has the lowest oil washing efficiency.

The oil washing efficiency of nonionic systems depends on the type of nonion. The OP-10 system has the highest oil washing efficiency. Amphoteric ions also have a greater impact on oil washing efficiency. The BS12 and CAB35 have the highest oil washing efficiency.

### 3.5. Study on the Performance of PGEP Compounded with Different Surfactants on Fracturing Fluid Flowback

#### 3.5.1. Effect of Different Cationic Surfactants on Drainage Efficiency

Experimental method: 3.5 g of PGEP, 3.5 g of cationic surfactant (1.75 g of 1831), 9 g of methanol, 17 g of BS12 betaine, and 67 g of water were taken and dissolved completely to determine their performance.

Figure 9a illustrates that the surface tension increases as the carbon chain of the cationic surfactant increases.

The interfacial tension, on the other hand, decreases and then increases. Figure 9b shows that after high temperature heating, the interfacial tension of the 1631 surface is reduced. However, the surface tension and interfacial tension of 1831 increase at the same time. 1631 has the best interfacial tension. Figure 9c shows that, in terms of drainage efficiency, it decreases with the increase of cationic surfactant carbon chains. 1231 has the highest drainage efficiency.

#### 3.5.2. Effect of Different Amphoteric Surfactants on Drainage Efficiency

Experimental method: 3.5 g of PGEP, 3.5 g of 1231, 9 g of methanol, 17 g of betaine, and 67 g of water were taken and dissolved completely to determine their performance.

Figure 10a shows that different betaines have a large effect on the surface/interfacial tension.

CAB35 was the most effective and had the lowest surface/interface tension. Figure 10b shows that after high temperature heating, the surface tension of LAB30 and CAB35 decreased by 5 mN/m. The interfacial tension of CAB35 and OAB40 was also reduced. Figure 10c illustrates that LAB30 and CAB35 are more effective in terms of drainage efficiency. CAB35 has the highest drainage efficiency and the best results.

#### 3.5.3. Effect of Different Alcohols on Drainage Efficiency

Experimental method: 3.5 g of PGEP, 3.5 g of 1231, 9 g of alcohol, 17 g of BS12, and 67 g of water were taken and dissolved completely to determine their performance.

Figure 11a shows that ethylene glycol is not able to make the solution transparent at the specified proportion.

The solution becomes clear only after the proportion of ethylene glycol is increased to 23% and the proportion of BS12 is increased to 25%. Figure 11b indicates that there is no change in the surface/interfacial tension of the system after heating at high temperatures. Alcohol had no significant effect on the temperature resistance of the system. Figure 11c illustrates that alcohols have little effect on the drainage efficiency of the system. Methanol and ethanol are more efficient in drainage.

Different betaines, cations, and alcohols have different effects on the drainage performance of the system. Surfactants with longer carbon chains have higher surface tension. The interfacial tension of surfactants has a suitable carbon chain length; too long or too short a carbon chain will increase the interfacial tension. 

## 4. Conclusions

In this study, PGEP was synthesized and evaluated for various properties after compounding with different surfactants. The following conclusions were obtained:The interfacial tension performance of PGEP after compounding with different surfactants can reach as low as 0.00034 mN/m. It can meet the technical requirements of interfacial tension (interfacial tension ≤ 5 × 10^−3^ mN/m) in the oilfield.The oil washing rate performance of PGEP after compounding with different surfactants is best up to 78.2%. It can meet the technical requirements of oil washing efficiency (Oil washing efficiency ≥ 40%) in an oilfield. Alcohols have little effect on the oil washing efficiency of the system.The surface tension of PGEP after compounding with different surfactants reaches as low as 27.42 mN/m, and the interfacial tension reaches as low as 0.21 mN/m. After high temperature (150 °C), the surface tension of PGEP after compounding with different surfactants reaches 22.35 mN/m, and the interfacial tension reaches 0.15 mN/m. The drainage efficiency of PGEP after compounding with different surfactants reaches 22%. It can meet the technical requirements of surface/interfacial tension and drainage efficiency (surface tension ≤ 30 mN/m, interfacial tension ≤ 3 mN/m; after high temperature (150 °C), surface tension ≤ 32 mN/m, interfacial tension ≤ 5 mN/m; drainage efficiency ≥ 15%) in the oilfield. The surface interfacial tension of the system remains constant after the concentration exceeds 0.2% and decreases with lower concentrations. The drainage efficiency increases with increasing concentrations in the range below 0.6%.

## Figures and Tables

**Figure 1 materials-17-00437-f001:**
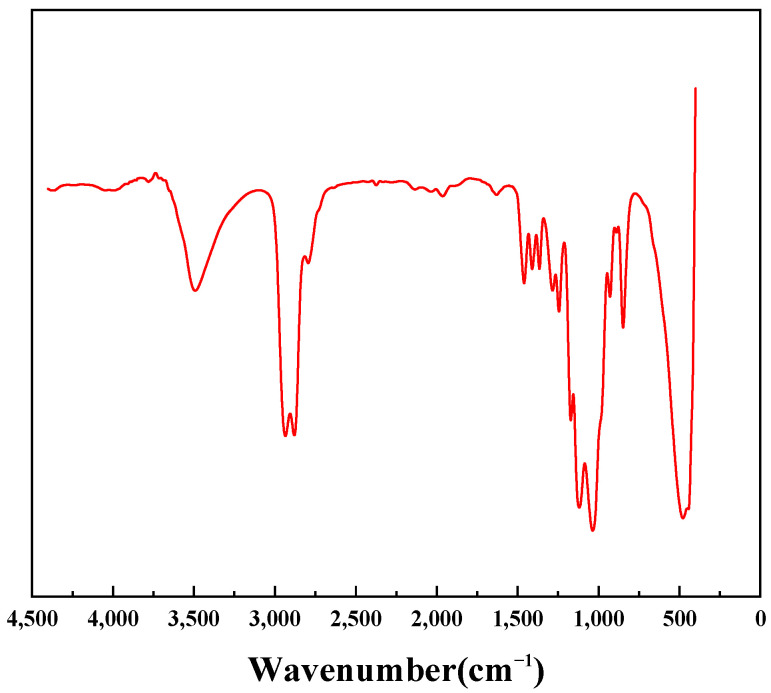
IR spectra of PGEP.

**Figure 2 materials-17-00437-f002:**
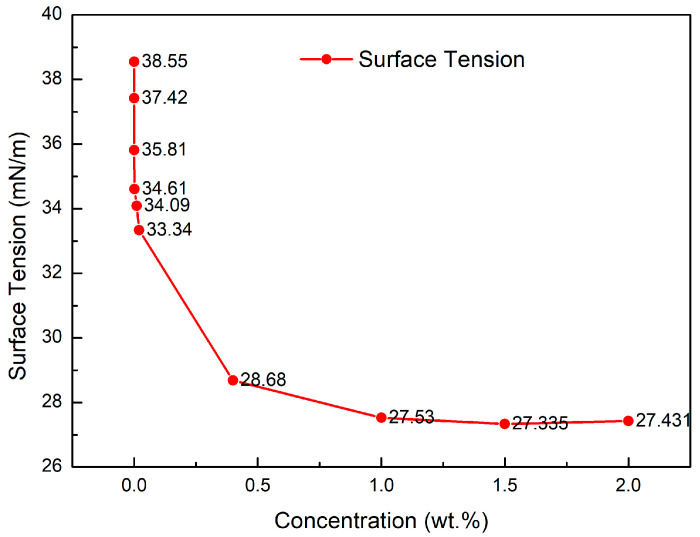
Surface tension at different concentrations of PGEP at 25 °C.

**Figure 3 materials-17-00437-f003:**
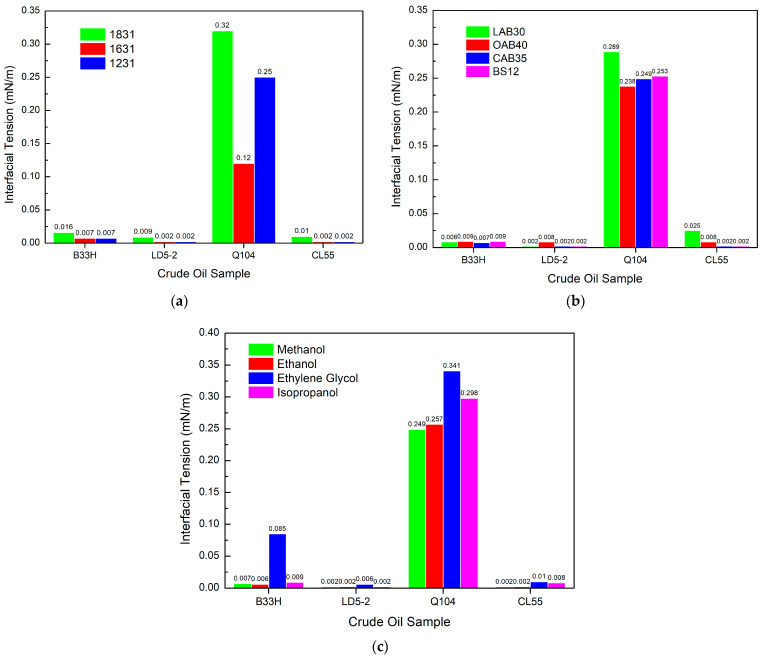
Effect of different components in cationic systems on interfacial tension: (**a**) cationic surfactants, (**b**) amphoteric surfactants, (**c**) alcohols.

**Figure 4 materials-17-00437-f004:**
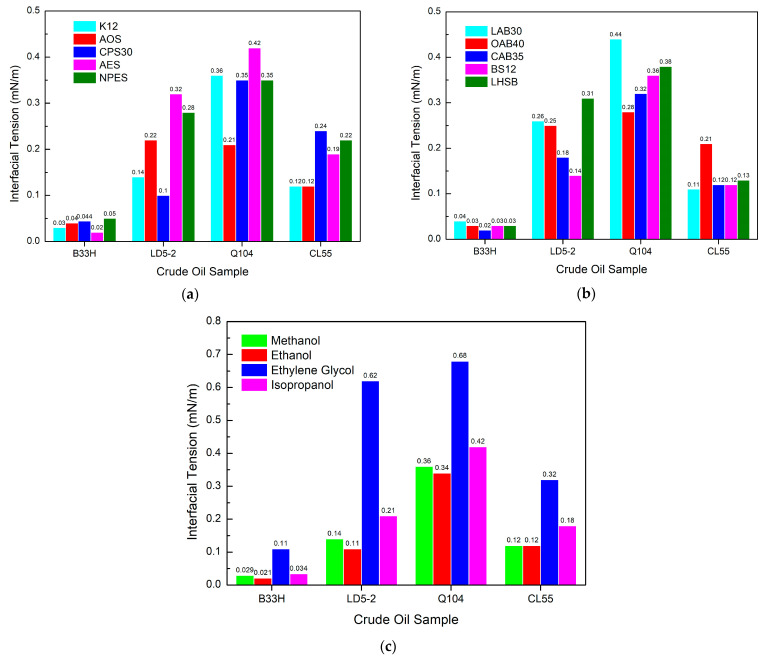
Effect of different components in anionic systems on interfacial tension: (**a**) anionic surfactants, (**b**) amphoteric surfactants, (**c**) alcohols.

**Figure 5 materials-17-00437-f005:**
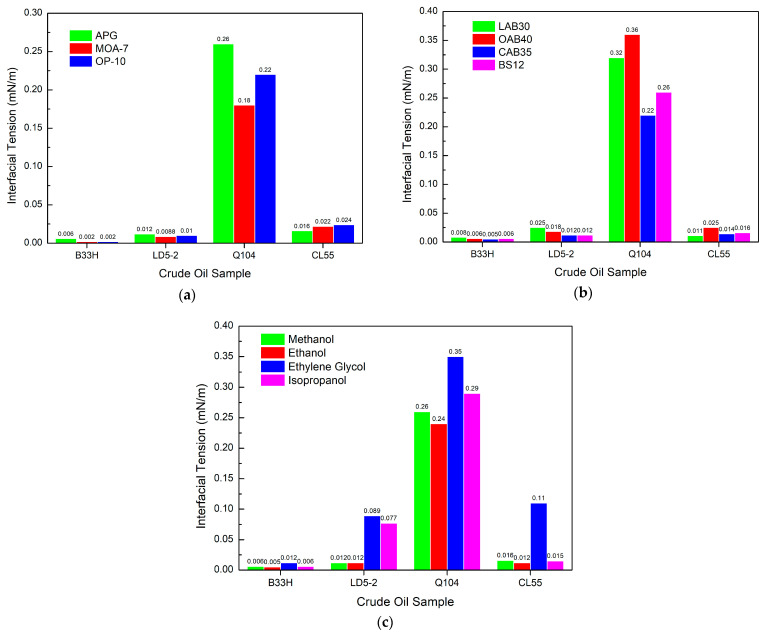
Effect of different components in nonionic systems on interfacial tension: (**a**) nonionic surfactants, (**b**) amphoteric surfactants, (**c**) alcohols.

**Figure 6 materials-17-00437-f006:**
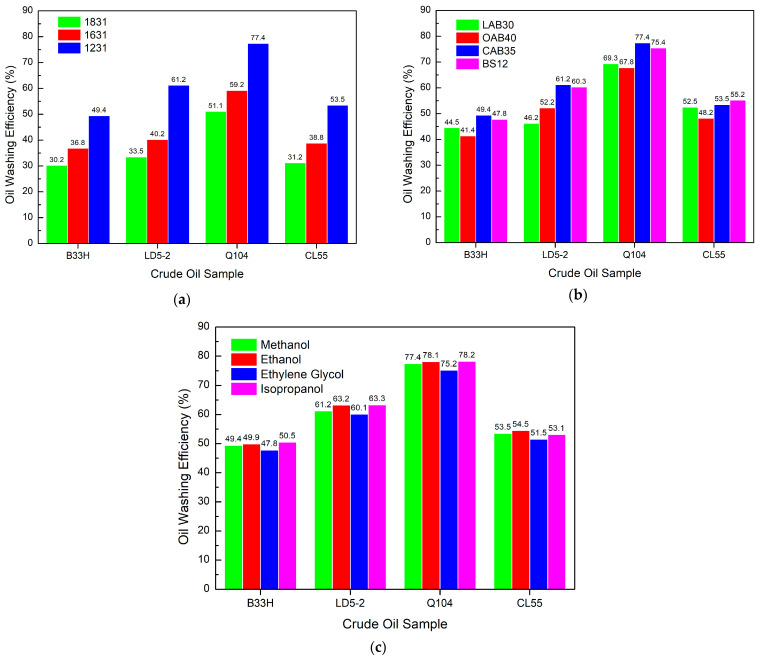
Effect of different components in cationic systems on oil washing efficiency: (**a**) cationic surfactants, (**b**) amphoteric surfactants, (**c**) alcohols.

**Figure 7 materials-17-00437-f007:**
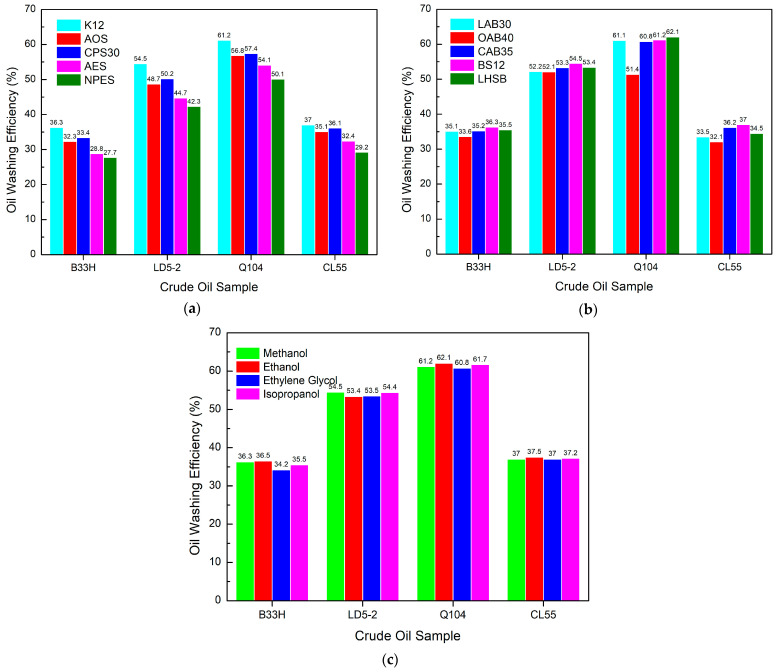
Effect of different components in anionic systems on oil washing efficiency: (**a**) anionic surfactants, (**b**) amphoteric surfactants, (**c**) alcohols.

**Figure 8 materials-17-00437-f008:**
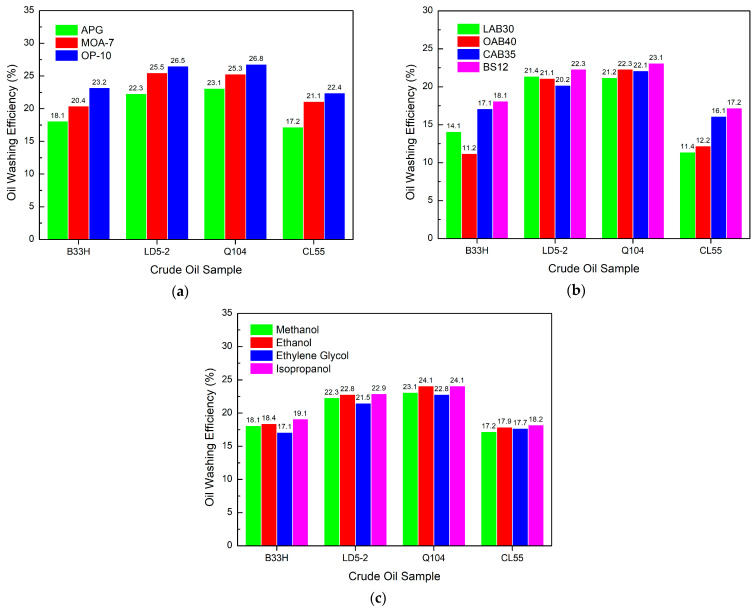
Effect of different components in a nonionic system on oil washing efficiency: (**a**) nonionic surfactants, (**b**) amphoteric surfactants, (**c**) alcohols.

**Figure 9 materials-17-00437-f009:**
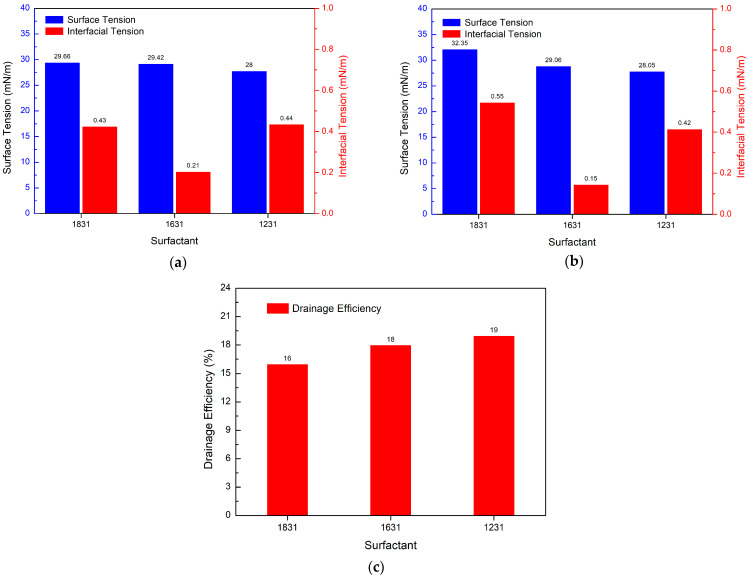
Effect of different cationic surfactants on the drainage efficiency: (**a**) surface/interfacial tension, (**b**) surface/interfacial tension after high temperature (150 °C), (**c**) drainage efficiency.

**Figure 10 materials-17-00437-f010:**
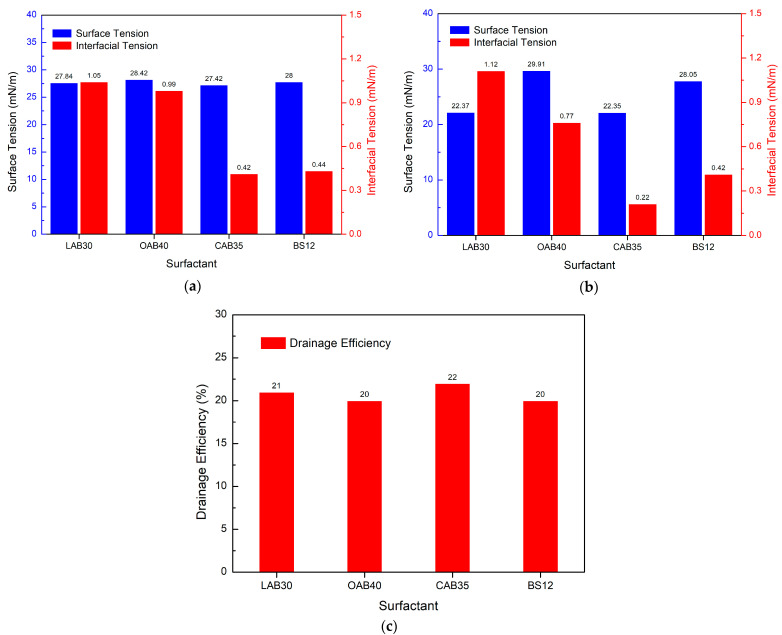
Effect of different amphoteric surfactants on the drainage efficiency: (**a**) surface/interfacial tension, (**b**) surface/interfacial tension after high temperature (150 °C), (**c**) drainage efficiency.

**Figure 11 materials-17-00437-f011:**
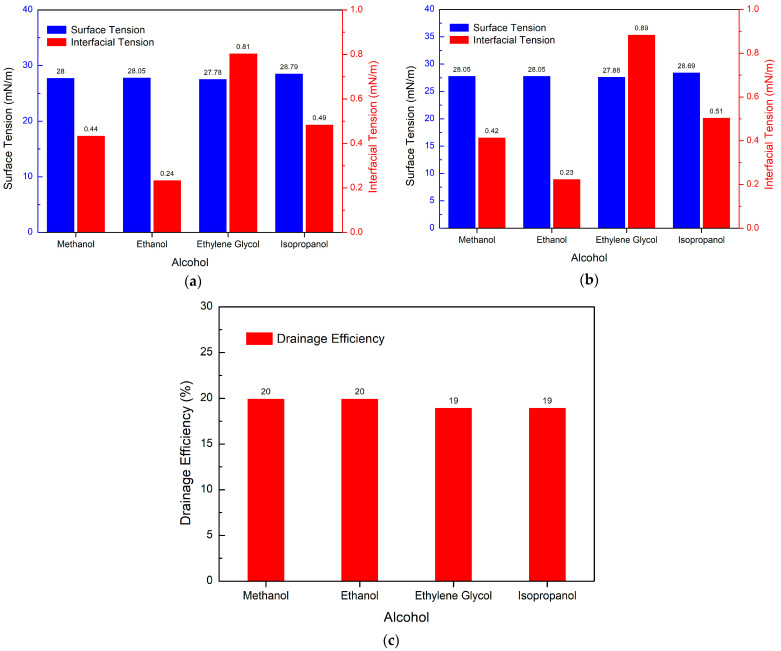
Effect of different alcohols on the drainage efficiency: (**a**) surface/interfacial tension, (**b**) surface/interfacial tension after high temperature (150 °C), (**c**) drainage efficiency.

## Data Availability

Data is contained within the article.

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
