# Peer review of "Study on the Performance Mechanism of Polyformaldehyde Glycol Ether Polymer for Crude Oil Recovery Enhancement"

_materials, 2024, doi:10.3390/ma17020437_

Round 1

Reviewer 1 Report

Comments and Suggestions for Authors

Thanks for the effort spent to put together the manuscript title:"Study on the Performance Mechanism of Polyformaldehyde Glycol Ether Polymers for Crude Oil Recovery Enhancement". The lack of novelty and poor presentation and English make the manuscript inappropriate for publication.

After careful reading of the manuscript, please find my comments as follow:

1. Clarify the novelty of your study comparing to others.

2. The introduction should be improved with more literature review to highlight the main objective of this paper.

3. Results are not presented clearly or comprehensively.  So, please discuss your results in more details to clarify them. 

4. Authors could add a line at the end of each subsection summarizing the graphs and the results they obtained, including their point of view.

5. This kind of results are considered a report not a scientific manuscript.

6- The manuscript needs to clear all figures.

7. In line 17 Correct PGEP are to PGEP is. 

8. In line 57: you mention " PGEP with different cationic surfactants, anionic surfactants, nonionic surfactants, and alcohols through oil washing efficiency and drainage efficiency" . You can replace it by "PGEP with different cationic , anionic , nonionic surfactants, and alcohols through oil washing efficiency and drainage efficiency" .

9. Figure 2. Provide the concentration used for this experiment to make it clear to the readers.

10. At what temperature did you do your experimental work and did you do the test at different temperature to see the affect of temperature on the chemical materials used in this study.

11. You compared the PGEP with FAPE in your manuscript. So, could you please gives more details to clarify how PGEP is more effective rather than FAPE.

12. In the conclusion you need to address the whole outcomes of this study.

Comments on the Quality of English Language

The English language of this manuscript needs to be improved.

Author Response

Reviewer: 1

Comments:

Thanks for the effort spent to put together the manuscript title: "Study on the Performance Mechanism of Polyformaldehyde Glycol Ether Polymers for Crude Oil Recovery Enhancement". The lack of novelty and poor presentation and English make the manuscript inappropriate for publication.

After careful reading of the manuscript, please find my comments as follow:

1. Clarify the novelty of your study comparing to others.

Response: Thank you very much for your suggestion. In this study, a new surfactant Polyformaldehyde Glycol Ether Polymers was synthesized. This product is currently not available in the market.

2. The introduction should be improved with more literature review to highlight the main objective of this paper.

Response: Thank you very much for your suggestion. The introduction has been improved with the addition of a more extensive literature review. However, because there are so few articles on PGEP, FAPE is used as a reference for comparison.

3. Results are not presented clearly or comprehensively.  So, please discuss your results in more details to clarify them. 

Response: Thank you very much for your suggestion. The experimental results have been discussed in more details.

4. Authors could add a line at the end of each subsection summarizing the graphs and the results they obtained, including their point of view.

Response: Thank you very much for your suggestion. Summaries of graphs and results have been added.

5. This kind of results are considered a report not a scientific manuscript.

Response: Thank you very much for your suggestion. In this study, a new surfactant was synthesized more environmentally friendly. The synthesized product was obtained as determined by infrared characterization and evaluated indoors.

6. The manuscript needs to clear all figures.

Response: Thank you very much for your suggestion. All figures have been reorganized. Specific data has been added to the bar charts for clearer understanding.

7. In line 17 Correct PGEP are to PGEP is. 

Response: Thank you very much for your suggestion. The error has been corrected.

8. In line 57: you mention " PGEP with different cationic surfactants, anionic surfactants, nonionic surfactants, and alcohols through oil washing efficiency and drainage efficiency". You can replace it by "PGEP with different cationic, anionic, nonionic surfactants, and alcohols through oil washing efficiency and drainage efficiency".

Response: Thank you very much for your suggestion. Replacements have been made as recommended.

9. Figure 2. Provide the concentration used for this experiment to make it clear to the readers.

Response: Thank you very much for your suggestion.

10. At what temperature did you do your experimental work and did you do the test at different temperature to see the effect of temperature on the chemical materials used in this study.

Response: Thank you very much for your suggestion. The surface/interfacial tension and oil washing efficiency were experimented at room temperature (25℃). The drainage efficiency was tested at room temperature (25℃) and 150°C in comparison.

11. You compared the PGEP with FAPE in your manuscript. So, could you please give more details to clarify how PGEP is more effective rather than FAPE.

Response: Thank you very much for your suggestion. FAPE is a commonly used surfactant in the oilfield, and its performance fully meets the needs of oilfield applications. PGEP is more environmentally friendly and green in the synthesis process, which is more in line with the requirements for chemicals in the oilfield.

12. In the conclusion you need to address the whole outcomes of this study.

Response: Thank you very much for your suggestion. The whole outcomes of this study have been listed in the conclusion as suggested.

Reviewer 2 Report

Comments and Suggestions for Authors

This manuscript presents a thorough investigation into the use of PGEP for enhancing crude oil recovery. The study is well-structured, and the clarity of the manuscript is commendable. I recommend acceptance of this manuscript contingent upon the authors' attention to the following concerns:

(1)   The absence of error bars in the experimental data need be addressed. Error bars are essential for understanding the variability and reliability of results, especially when the experiments yield similar outcomes, notably in Figure 7.

(2)   The manuscript would benefit from the inclusion of control experiments for interfacial tension measurements. Specifically, experiments without the addition of alcohol and with various cationic, anionic, and nonionic surfactants should be presented to establish a baseline for comparison.

(3)   Given that the authors propose PGEP as a substitute for FAPE, it would be valuable to either discuss or, ideally, conduct comparative experiments with FAPE. This would provide a direct performance benchmark and further substantiate the manuscript's claims.

Author Response

Reviewer: 2

Comments:

This manuscript presents a thorough investigation into the use of PGEP for enhancing crude oil recovery. The study is well-structured, and the clarity of the manuscript is commendable. I recommend acceptance of this manuscript contingent upon the authors' attention to the following concerns:

 (1)   The absence of error bars in the experimental data need be addressed. Error bars are essential for understanding the variability and reliability of results, especially when the experiments yield similar outcomes, notably in Figure 7.

Response: Thank you very much for your suggestion. Specific data has been added to the bar charts for clearer understanding.

 (2)   The manuscript would benefit from the inclusion of control experiments for interfacial tension measurements. Specifically, experiments without the addition of alcohol and with various cationic, anionic, and nonionic surfactants should be presented to establish a baseline for comparison.

Response: Thank you very much for your suggestion. Surfactants need to be used in the form of compounding in the oilfield, which can achieve the result of using less quantity with better effect. The main objective of this study is to evaluate the effectiveness of PGEP compounded with various surfactants for oilfield application.

 (3)   Given that the authors propose PGEP as a substitute for FAPE, it would be valuable to either discuss or, ideally, conduct comparative experiments with FAPE. This would provide a direct performance benchmark and further substantiate the manuscript's claims.

Response: Thank you very much for your suggestion. FAE is a commonly used surfactant in the oilfield, and its performance fully meets the needs of oilfield applications. PGEP is more environmentally friendly and green in the synthesis process, which is more in line with the requirements for chemicals in the oilfield. It can be used as a surfactant instead of FAE in oilfield development.

Reviewer 3 Report

Comments and Suggestions for Authors

In this paper authors investigate the synthesis of PGEP and evaluated for various properties after compounding with different surfactants.

The authors shows that the interfacial tension performance of PGEP after compounding with different surfactants can reach as low as 0.00034mN/m, so the oil washing rate performance of PGEP after compounding with different surfactants is best up to 78.2%.

I consider that the paper is clear, the figures are good. The work may be published in the form presented.

Author Response

Reviewer: 3

Comments:

In this paper authors investigate the synthesis of PGEP and evaluated for various properties after compounding with different surfactants.

The authors shows that the interfacial tension performance of PGEP after compounding with different surfactants can reach as low as 0.00034mN/m, so the oil washing rate performance of PGEP after compounding with different surfactants is best up to 78.2%.

I consider that the paper is clear, the figures are good. The work may be published in the form presented.

Response: Thank you very much for your suggestion.

Round 2

Reviewer 1 Report

Comments and Suggestions for Authors

Thank you for addressing the whole comments and provide the requirement details needed.